# Stomatal responses of differently CO₂-acclimated plants to natural and experimental CO₂ gradients

**Peter Manuel Kammer** [1]*, **Dominik Lukas Wermelinger** [1,2], **Janick Michael Klossner**[1], **Jonathan Simon Steiner**[1], **Christian Schöb**[3]

**1** PH Bern, NT Biologie, Bern, Switzerland, **2** Physics Institute, University of Bern, Bern, Switzerland, **3** Biology, Geology, Physics and Inorganic Chemistry, Área de Biodiversidad y Conservación, Universidad Rey Juan Carlos, Móstoles, Spain

* peter.kammer@phbern.ch (PK)

## Abstract

Stomata, the microscopic pores in the epidermis of plant leaves, facilitate gas exchange between the leaf interior and the external environment and thus play a pivotal role in the global carbon, water, and energy cycles. Their development and function are influenced by atmospheric CO₂ levels. However, the mechanisms and the extent of this influence remain incompletely understood. Understanding the stomatal responses to variations in atmospheric CO₂ is essential for predicting their impact on future plant productivity or the role of vegetation as a carbon sink. We investigated the stomatal responses – specifically stomatal aperture and stomatal frequency – of C₃ plants acclimated to either ambient (approximately 40 Pa) or reduced (approximately 30 Pa) partial pressure of CO₂ (pCO₂). Stomatal responses were assessed both under natural conditions along an altitudinal gradient and under controlled experimental conditions with defined CO₂ levels (30 vs. 42 Pa). Our results showed that, under experimental conditions, stomatal frequency decreased with increasing CO₂ levels in both plant groups, regardless of their acclimation history. Furthermore, plants exposed to reduced pCO₂ exhibited smaller stomatal apertures compared to those grown at ambient pCO₂. Above-ground fresh biomass was higher at ambient CO₂ levels than at reduced levels. Interestingly, no significant difference in stomatal frequency was observed between plants that were acclimated and grown under reduced pCO₂ (at 2,970 m a.s.l.) and those under ambient pCO₂ (540 m). These results support the hypothesis that the inverse stomatal response to CO₂ – i.e., a reduction in stomatal frequency with increasing CO₂ levels – is a general physiological pattern in C₃ plants in the range of 30–42 Pa. However, this response pattern can be overridden by specific (locally occurring) environmental conditions such as low humidity or elevated temperatures. Furthermore, the findings are consistent with the optimal stomatal theory, which postulates that plants regulate stomatal conductance, including stomatal frequency and aperture, to maximize photosynthesis while minimizing water loss to achieve optimal water-use efficiency.

**Data availability statement:** All data files are available from the open Zenodo database (accession number(s) DOI: 10.5281/zenodo.17752070).

**Funding:** The author(s) received no specific funding for this work.

**Competing interests:** The authors have declared that no competing interests exist.

## Introduction

Stomata are microscopic pores located in the leaf epidermis, each bordered by a pair of guard cells, that facilitate gas exchange between the leaf interior and the external environment. Water vapor is released through the stomatal opening, driving the movement of water from the soil through the plant into the atmosphere [1]. At the same time, carbon dioxide ($CO_2$), the primary substrate of the dark reaction of photosynthesis, diffuses along its concentration gradient through the stomata into the leaf [1]. Given the substantial gas fluxes through stomata, the conductance of these pores (stomatal conductance) plays a crucial role in the global carbon, water, and energy cycles, thereby influencing the global climate [2–6]. Stomatal conductance responds to various environmental factors, including atmospheric $CO_2$ concentration [7]. With continuously increasing atmospheric $CO_2$ levels due to the combustion of fossil fuel [8], understanding the response of stomatal conductance to variations in $CO_2$ levels is critical for predicting its impact on future plant productivity or the role of vegetation as a carbon sink.

In general, plants exhibit a decrease in stomatal conductance in response to elevated $CO_2$ levels, a pattern extensively documented in multiple meta-analyses [7,9–12]. This response is commonly explained by the optimal stomatal theory, which postulates that plants regulate stomata to maximize photosynthesis and minimize water loss to achieve optimal water-use efficiency [7,13,14]. However, under specific environmental conditions such as arid and warm environments or during periods of drought, plants may exhibit an increase in stomatal conductance in response to elevated $CO_2$ levels [15], indicating that the inverse relationship between stomatal conductance and $CO_2$ levels is not universal [16].

Stomatal conductance is determined by three components: stomatal aperture (hereafter "SA"), defined as the width of the stomatal pore; stomatal size, referring to the length of the stomatal pore; and stomatal density (hereafter "SD"), which represents the number of stomata per unit leaf area [17–20]. The alteration of SA constitutes a short-term, reversible, and immediate response of plants to fluctuations in environmental conditions [4,10,16]. Generally, an increase in $CO_2$ levels induces a reduction in stomatal pore width, whereas a decrease in $CO_2$ levels leads to its expansion [21–24]. In contrast to changes in SA, variations in stomatal size and SD reflect the longer-term phenotypic plasticity of plants in response to environmental shifts [4,10,16]. Stomatal pore length does not exhibit significant sensitivity to changes in $CO_2$ levels [19,25,26]. Unlike SA and size, the response of SD to $CO_2$ levels remains unclear. A reduction in SD is generally considered to be the common response to elevated $CO_2$ levels [16,22,24,26]. However, Royer [27] reported that only about 40% of 127 stomatal responses across 68 species exhibited an inverse relationship with atmospheric $CO_2$, whereby SD declined with increasing $CO_2$ levels and increased with decreasing $CO_2$ levels. The remaining responses were either positive or not statistically significant. Similarly, Poorter et al. [12], who evaluated 400 experimental stomatal responses from 120 species, concluded that SD is only marginally responsive to variation in atmospheric $CO_2$ levels. The datasets analyzed by Royer [27] and Poorter et al. [12] originated from experiments conducted under controlled conditions in enclosed growth chambers, greenhouses, and open-top chambers (OTCs).

Most studies have examined SD responses to $CO_2$ levels exceeding the present-day concentrations, in some cases up to 3200 ppm [12], i.e., $CO_2$ concentrations which have not been observed in the last 400 million years [28], and to which plants have therefore not been exposed for millennia [27]. Additionally, angiosperms often exhibit little to no SD response to increases in $CO_2$ levels above 400 ppm, likely due to their evolutionary origin and diversification under low atmospheric $CO_2$ conditions, i.e., below 400 ppm [25–27,29,30]. Consequently, investigating SD responses to $CO_2$ variations experienced by plants over the past millennia may provide more reliable insights [31]. The altitudinal gradient is one such natural framework for $CO_2$ variations [26,32], reflecting the $CO_2$ conditions experienced by plants during downward and upward migration in glacial and interglacial periods.

However, similar to the response of SD to variations in atmospheric $CO_2$ levels, the SD response to altitude – and consequently to differences in $CO_2$ partial pressure (hereafter "pCO$_2$") – remains inconsistent. Several studies have reported a significantly positive correlation between altitude and SD [33–42], supporting the inverse relationship between SD and pCO$_2$. The observed increase in SD with altitude is interpreted as an adaptive mechanism to mitigate the reduced photosynthetic potential of plants growing at higher elevations, which results from the decline in pCO$_2$ [39].

Conversely, other studies have identified a significantly negative relationship between altitude and SD [37,43–46]. Thus, the response of SD, as well as stomatal conductance in general, to changes in atmospheric $CO_2$ remains incompletely understood [11,15,46].

An altitudinal gradient is invariably accompanied by changes in environmental factors beyond pCO$_2$ [32,39,47], which can also influence SD, such as irradiance (including UV-B radiation), humidity, and air temperature [5,6,20,24,26,48]. The inconsistency in stomatal responses to changes in pCO$_2$ could, therefore, be partially attributed to such environmental factors [49]. Consequently, to observe an undisturbed response of SD to changes in atmospheric $CO_2$, all other environmental factors must be maintained as constant as possible.

In this study, we tested plants originating from populations that have persisted for extended periods at either high altitudes (hereafter "high-altitude taxa") or low altitudes (hereafter "low-altitude taxa"). These plants were grown in controlled chambers under two $CO_2$ treatments: a reduced pCO$_2$ of 30 Pa, reflecting conditions at the native elevation of the high-altitude taxa (2,970 m a.s.l.), and an ambient pCO$_2$ of 42 Pa, corresponding to conditions at 540 m a.s.l. This setup effectively constituted a reciprocal transplantation experiment. Other environmental factors, including air temperature, vapor pressure deficit (hereafter "VPD"), and irradiance, were kept as equal as possible in the two growth chambers to minimize the confounding effects of those environmental factors in the experiment.

In addition to assessing the stomatal frequency (hereafter "SF") of plants cultivated in growth chambers, we quantified SF of plants of the same taxa growing at their natural sites along a natural $CO_2$ gradient.

SF was defined to encompass SD and the stomatal index (hereafter "SI"), which is calculated as the ratio of stomata to the total number of epidermal cells plus stomata [17,26]. In contrast to SD, SI is not influenced by variations in epidermal cell size [27,39,45].

Following the optimal stomatal theory, we hypothesized that (H1) SA of plants grown in growth chambers under reduced pCO$_2$ would be significantly larger compared to those grown in chambers under ambient pCO$_2$; (H2) SF of high-altitude taxa growing at their natural sites would be significantly higher than that of low-altitude taxa growing at their natural sites; (H3) SF of plants cultivated under ambient pCO$_2$ would be significantly lower than that of plants grown under reduced pCO$_2$; and that (H4) the biomass of plants cultivated under ambient pCO$_2$ would be greater than that of plants grown under reduced pCO$_2$, as water use efficiency is higher at ambient $CO_2$ due to lower stomatal conductance.

Moreover, we examined the simultaneous effects of multiple environmental factors, including pCO$_2$, air temperature, VPD, and irradiance, on SA and SF. To this end, we incorporated additional data from Kammer et al. [50]. Based on existing literature [5,6,20,24,26,48], we hypothesized that SA would increase with rising temperature and irradiance but decrease with increasing VPD (H5). Additionally, we expected that SF would decrease with increasing temperature and VPD while increasing with higher irradiance (H6).

## Materials and methods

### Plant material

The objective of this study was to investigate the stomatal response to variations in $CO_2$ in plants acclimated to ambient $CO_2$ levels compared to plants acclimated to reduced $CO_2$ levels. For the latter, we focused on plant populations that have been growing at high altitudes for extended periods. To this end, we selected *Arabis alpina* (hereafter "*A. alpina*") as the high-altitude counterpart of *Arabidopsis thaliana* (hereafter "*A. thaliana*"). Both species, *A. thaliana* and *A. alpina*, belong to the plant family Brassicaceae. Additionally, we selected a second pair of taxa from the legume family (Fabaceae), namely *Anthyllis vulneraria* subsp. *carpatica* (hereafter "*A. carpatica*"), representing low-altitude taxa, and *Anthyllis vulneraria* subsp. *valesiaca* (hereafter "*A. valesiaca*"), representing its high-altitude counterpart.

In Switzerland, *A. thaliana* occurs at altitudes ranging from 200 to 2,000 m a.s.l. (S1 Fig), with a median elevation of 453 m, whereas *A. alpina* is found between 300 and 3,250 m a.s.l. (median = 1,629 m). Similarly, *A. carpatica* is found between 200 and 2,800 m a.s.l. (median = 725 m), while *A. valesiaca* occurs between 200 and 3,100 m a.s.l. (median = 1,882 m). All of this data were obtained from the database of the National Data and Information Center on the Swiss Flora (www.infoflora.ch).

Leaves for SF analysis of *A. alpina* and *A. valesiaca* were collected at two sites in the Swiss Alps at an elevation of 2,970 m a.s.l. (S1 Table), while those of *A. thaliana* and *A. carpatica* were collected at 540 m a.s.l. Seeds used for the growth chamber experiments were collected from the same field locations, except for *A. thaliana*, whose seeds were obtained from a laboratory-maintained Columbia (Col-0) wild-type lineage to enable comparison with previously published results. All plants from which leaves and seeds were collected grew in open, fully sun-exposed habitats that could be affected by drought at times.

### Plant cultivation

Plants were cultivated in two custom-made growth chambers (Astromec, Muri b. Bern, Switzerland). The control chamber was used for experiments under ambient pressure, whereas in the treatment chamber, air pressure and consequently $pCO_2$ were reduced. To avoid any potential chamber effects, the designation of control and treatment chambers was alternated in each experiment. The chambers had a volume of 0.21 m³, featuring a transparent topside for illumination, a transparent front door for monitoring, and two fittings for tubes on opposing sidewalls. A vacuum pump (Seco Tiny SV 1003 A; Busch, Maulburg, Germany) operating on the rotating vane principle was connected to the outflow tube, while a needle valve (B-1RF4; Arbor-Swagelok, Niederrohrdorf, Switzerland) was installed on the inflow tube. The vacuum pump continuously operated at a suction capacity of 3 m³ h⁻¹, ensuring complete air renewal in the chambers every five minutes.

Air pressure in the treatment chamber was reduced by adjusting the inflow tube valve, leading to an outflow exceeding the inflow. The inflow tube valve of the control chamber was fully opened, maintaining ambient pressure conditions. Air pressure (GPB 2300; Greisinger, Regenstauf, Germany), temperature and relative air humidity (Humicap HM70; Vaisala, Helsinki, Finland), along with $CO_2$ concentration (Carbocap GM70; Vaisala, Helsinki, Finland), were continuously monitored within the chambers. To calculate the vapor pressure deficit (VPD), the saturation vapor pressure was approximated by applying the Magnus formula [51]. Evaporation from an open water surface ensured that the relative humidity in both chambers was largely identical.

Each chamber was illuminated by two LED panels (Led Fox, Reinach, Switzerland), each equipped with 66 diodes (3 W each) emitting at four different wavelengths (18 x 450 nm, 6 x 610 nm, 6 x 630 nm, 36 x 660 nm). An even distribution of light within the chambers was achieved by placing four layers of white wrapping paper on top of the chambers. Light intensity at the plant level during the 12-hour photoperiod was 175 mol quanta PPFD m⁻² s⁻¹ (LI-190SA Quantum Sensor; LI-COR, Lincoln NE, U.S.A.). As temperature and humidity were not actively regulated, these factors fluctuated in accordance with ambient conditions.

Plants were hydroponically grown in individual meshed plastic pots (35 mm lower and 50 mm upper diameter, 50 mm height) filled with expanded clay pellets (diameter 2–6 mm). Following germination under ambient pressure conditions, the seedlings with two cotyledons were transferred to the growth chambers. Because the *Anthyllis* subspecies developed more rapidly than *A. alpina* and *A. thaliana*, they were cultivated in the growth chambers for 28 days, whereas the latter species were grown for 35 days. Plants were arranged in a fixed grid within the growth chambers, and their positions were not altered during the experiments to prevent any damage to the root system. The nutrient solution was replaced on days 14 and 28 to prevent nutrient depletion. The composition of the nutrient solution is provided in S2 Table.

## Measurements of plant traits

For measuring stomatal aperture (SA), one fully developed leaf per plant was excised and immersed in liquid nitrogen immediately after the growth chambers were opened at the end of a growth period. The leaves were then stored at −20 °C until SA was measured using an Olympus BX51 microscope, an Olympus SC50 digital microscope camera, and CellSens Entry (Version 1.14) imaging software (all components Olympus, Tokyo, Japan). Ten SA per leaf were measured to the nearest 0.01 µm.

For measuring stomatal frequency (SF), only leaves that had developed from newly formed buds after seedling transfer into the growth chambers were used. Preliminary analyses indicated that stomata were largely absent on the adaxial leaf surfaces [50]. Therefore, only SF on the abaxial surface was assessed. For this purpose, impressions of the abaxial surfaces of two fully developed leaves per plant were taken using dental impression gel consisting of Xantopren VL Plus and Optosil P Plus (Heraeus Kulzer, Hanau, Germany) [52,53]. The impressions were consistently taken from the same leaf region: adjacent to the midvein, and midway between the tip and the base of a leaf or, in the case of the *Anthyllis* subspecies, of a terminal foliole. Transparent, positive imprints were subsequently prepared on microscope slides from the gel molds using standard nail varnish [52,53]. Stomata and epidermal cells were counted in two or three sections per leaf using the same optical system as described above. For *A. thaliana* and *A. alpina*, counts were conducted in sections of 0.04 mm$^2$ (200 µm x 200 µm). Due to the larger epidermal cells of the *Anthyllis* subspecies, stomata and epidermal cells were counted in sections of 0.09 mm$^2$ (300 µm x 300 µm) for *A. carpatica* and *A. valesiaca*.

To determine SF in plants growing at their natural sites, one leaf from ten individuals per taxon was randomly sampled and transported to the laboratory, ensuring continuous moisture retention. Subsequently, stomata and epidermal cell counts were conducted as described above.

Preliminary analyses revealed a strong positive correlation between shoot fresh weight and shoot dry weight ($r = 0.975$, $n = 130$; [50]). Additionally, the shoot/root ratio remained unaffected by changes in $pCO_2$ [50]. Therefore, only above-ground fresh weight was determined by excising leaf rosettes from the roots and weighing them to the nearest 0.1 mg immediately after the end of the experiment.

## Statistical analyses

To test hypothesis H1, a linear mixed model was applied with stomatal aperture ($n = 800$) as the response variable, and origin (high vs. low altitude), taxon (*Arabis/Arabidopsis* vs. *Anthyllis*), $pCO_2$ treatment (30 vs. 42 Pa) and their interactions as the main effects. Chamber ID was included as a random effect to control for potential chamber specific effects.

To test H2, linear models were used with stomatal density (SD; $n = 82$) and stomatal index (SI; $n = 82$) as response variables and origin, taxon and their interactions as main effects.

In addition, SD ($n = 239$ and 262 for low and high altitude, respectively) and SI ($n = 239$ and 254 for low and high altitude, respectively) were compared between plant individuals sampled in situ and those cultivated under experimental $pCO_2$ conditions corresponding to their altitude of origin. For this purpose, a linear model was used with SD and SI as response variables, and taxon, treatment and their interaction as main effects.

To test H3, linear mixed models were applied with SD (n = 836, recorded at 21 positions in 2 growth chambers), SI (n = 828, 21 positions in 2 chambers) and epidermal cell density (n = 828, 21 positions in 2 chambers) as response variables, and origin, taxon, $pCO_2$ treatment and their interactions as main effects. Position nested within chamber ID were included as random terms. The term "position" refers to the specific location within the grid in the growth chambers where an individual plant grew. It was included in the models to account for potential microenvironmental variation associated with spatial placement within the chamber.

To test H4, we conducted a linear mixed model with fresh weight of the aboveground biomass (n = 194, recorded in 14 chambers of 7 experiments) as response variable, and taxon, origin, $pCO_2$ treatment and their interactions as main effects. Chamber nested within experiment were included as random terms.

To test H5 and H6, linear mixed models were used with SA (n = 1200, across 30 leaves per experiment in 4 experiments), SD (n = 1926, on 791 leaves of 296 individuals in 9 experiments) and SI (n = 490, on 273 leaves of 177 individuals in 6 experiments) as response variables, and $pCO_2$, air temperature, VPD, irradiance and their interactions with species (*A. thaliana* vs. *A. alpina*) as main effects. Leaf nested within individual nested within experiment were included as random terms.

Linear mixed models were conducted in R version 4.3.0 using the lme() function of the nlme package [54]. For linear models we used the lm() function of the base package [55].

For post hoc tests, we used emmeans() of the emmeans package [56] to compute estimated marginal means and cld() of the multcomp package [57] for pairwise comparisons using the tukey method and the sidak method for the adjustment of significance.

To assess potential observation bias, observer identity was included as a covariate in models that involved multiple data collectors. This analysis revealed no significant influence of the observer on the collected data.

### Use of artificial intelligence tools

Artificial intelligence was used exclusively to improve the linguistic quality of the manuscript. The authors wrote the manuscript in English and entered the sections they considered inadequate into the AI tool (ChatGPT, OpenAI) to obtain alternative formulations in scientific language. All AI-generated suggestions were carefully reviewed by the authors. Only those AI-generated outputs that accurately reflected the original intended meaning and that improved the clarity or readability of the text were partially or fully incorporated into the final manuscript.

## Results

### Response of stomatal aperture to variations in $pCO_2$

The response of stomatal aperture (SA) to variations in $pCO_2$ was taxon-dependent, as indicated by a significant interaction between taxon and treatment (S3 Table). As shown in Fig 1, the two *Anthyllis* subspecies as well as *A. alpina* exhibited a reduction in SA at 30 Pa $pCO_2$ compared to 42 Pa $pCO_2$, whereas *A. thaliana* displayed the opposite response. However, post hoc tests did not detect statistically significant pairwise differences between treatments. Furthermore, the high-altitude taxa generally exhibited significantly smaller SA than their low-altitude counterparts.

### Response of stomatal aperture to different environmental variables

SA in both species exhibited the strongest correlation with vapor pressure deficit (VPD; Table 1), with increasing VPD leading to stomatal closure (Fig 2C). This response was slightly species-specific, as indicated by a significant interaction term (Table 1), with *A. alpina* showing a more pronounced reduction in SA than *A. thaliana*. Additionally, irradiance had a species-specific effect on SA. Under low irradiance (85 µmol m$^{-2}$ s$^{-1}$), both species displayed similar SA. However, at higher irradiance (175 µmol m$^{-2}$ s$^{-1}$), *A. thaliana* exhibited larger SA compared to *A. alpina* (Fig 2D). The statistical analysis revealed no significant effects of $pCO_2$ or temperature on SA.

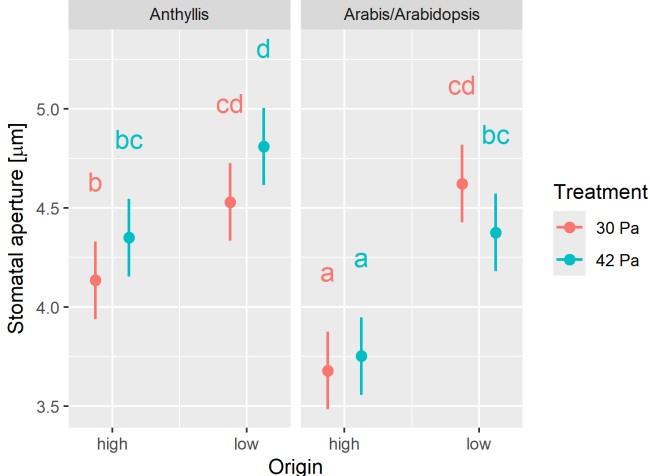

**Fig 1. Stomatal aperture response to variations in pCO₂.** Stomatal aperture (SA; n = 800) was measured in plant taxa from different altitudinal origins: high-altitude taxa ("high"; altitude of origin 2,970 m a.s.l.), including *Anthyllis vulneraria* subsp. *valesiaca* and *Arabis alpina*, and low-altitude taxa ("low"; 540 m a.s.l.), including *Anthyllis vulneraria* subsp. *carpatica* and *Arabidopsis thaliana* (Col-0). All plants were cultivated under reduced (30 Pa) and ambient pCO₂ (42 Pa). Marginal means along with their 95% confidence intervals are presented. Statistical analysis results are provided in S3 Table. Different letters denote statistically significant differences at *p* < 0.05, as determined by post hoc tests.

**Table 1. Marginal effects of environmental factors on stomatal aperture.**

| Stomatal aperture (SA); n = 1,200 | Chisq | Df | Pr(>Chisq) |
|---|---|---|---|
| (Intercept) | 0.0430 | 1 | 0.83581 |
| Species | 1.2747 | 1 | 0.25890 |
| pCO₂ | 3.2612 | 1 | 0.07094 |
| Species × pCO₂ | 0.3281 | 1 | 0.56677 |
| Temperature | 1.2520 | 1 | 0.26316 |
| Species × Temperature | 3.0578 | 1 | 0.08035 |
| VPD | 37.9550 | 1 | 7.24e-10*** |
| Species × VPD | 4.7007 | 1 | 0.03015* |
| Irradiance | 7.0576 | 1 | 0.00789** |
| Species × Irradiance | 8.4439 | 1 | 0.00366** |
| Irradiance x pCO₂ | 7.9687 | 1 | 0.00476** |
| Species x Irradiance x pCO₂ | 2.2724 | 1 | 0.13170 |

Type-III ANOVA of the linear mixed model testing the statistical significance of partial pressure of CO₂ (pCO₂), air temperature, vapor pressure deficit (VPD) and irradiance as main effects for stomatal aperture (SA) of *Arabis alpina* and *Arabidopsis thaliana* (Col-0).

Furthermore, SA was influenced by the interaction between irradiance and pCO₂ (Table 1). Under low irradiance, SA increased with rising pCO₂. However, at high irradiance, where SA was generally larger, no further variation was observed in response to changes in pCO₂.

## Stomatal frequency of plants growing at their natural sites

Stomatal density (SD) and stomatal index (SI) exhibited significant taxon- and origin-specific variation, as indicated by significant interaction terms (S4 Table). While the two *Anthyllis* subspecies displayed no significant differences in stomatal

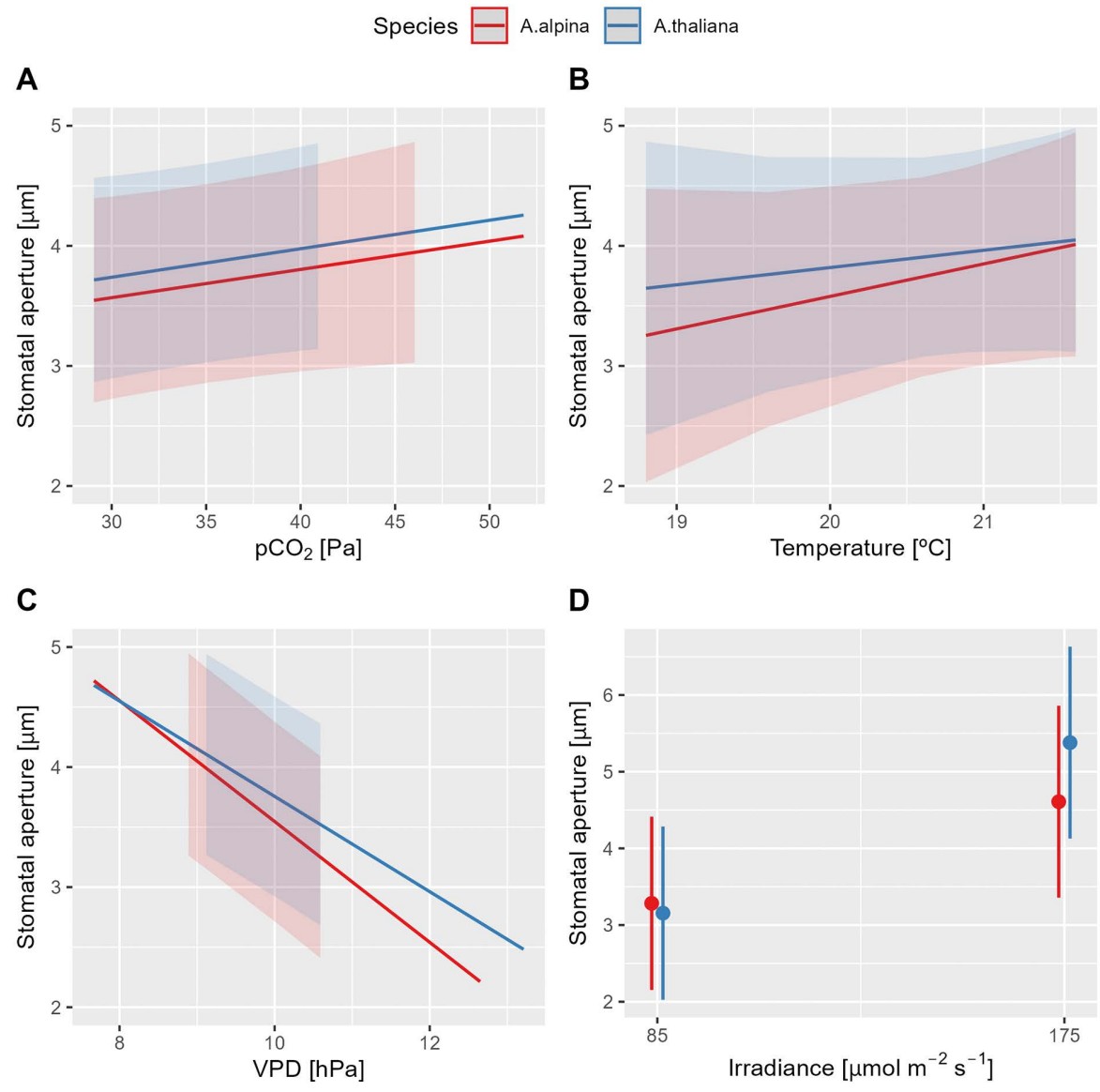

**Fig 2. Marginal effects of environmental factors on stomatal aperture.** Effects of (A) partial pressure of $CO_2$ ($pCO_2$), (B) air temperature, (C) vapor pressure deficit (VPD), and (D) irradiance on stomatal aperture (SA; n = 1'200) of *Arabis alpina* and *Arabidopsis thaliana* (Col-0). The data were obtained from four experiments conducted under environmental conditions that varied within the following minimum and maximum ranges: $pCO_2$ (29.1–51.8 Pa), temperature (18.8–21.6°C), VPD (7.7–13.2 hPa), and irradiance (85 and 175 µmol m$^{-2}$ s$^{-1}$). Marginal means along with their 95% confidence intervals are presented. Statistical analysis results are provided in Table 1.

frequency (SF) when growing at their natural sites, *A. thaliana* at 540 m a.s.l. had significantly higher SD and SI than its high-altitude counterpart, *A. alpina* growing at 2,970 m a.s.l. (Fig 3).

### Stomatal frequency of plants from natural sites and their laboratory-grown conspecifics exposed to altitude-specific $pCO_2$ levels

SD did not differ significantly between plants cultivated under altitude-specific $pCO_2$ conditions and those growing at their natural sites (Figs 4A and 4C). Similarly, no significant difference was observed in the SI of high-altitude taxa (Fig 4B).

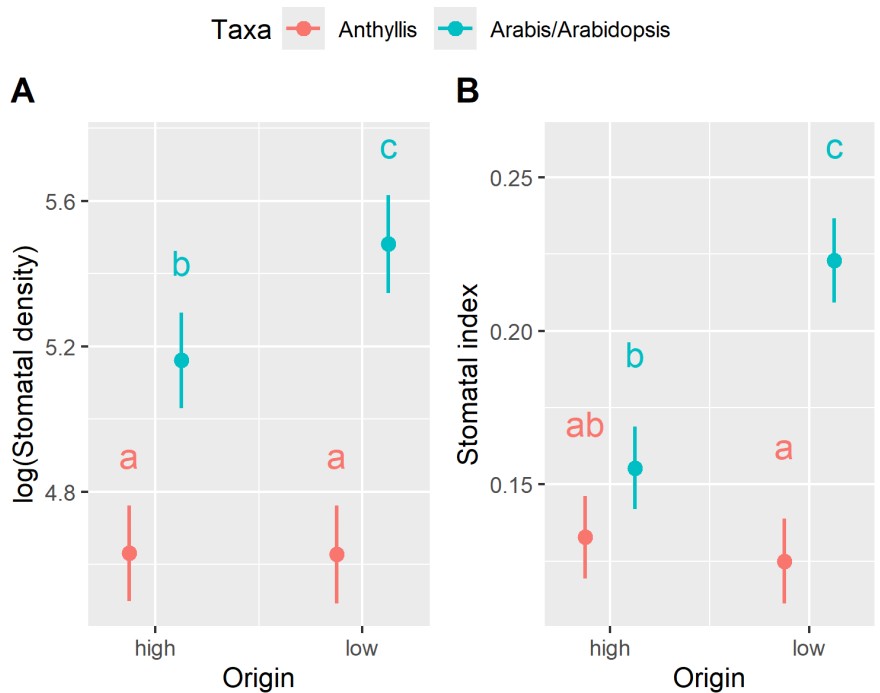

**Fig 3. Stomatal frequency of plants growing at their natural sites. (A)** Stomatal density (SD; n = 82) and **(B)** stomatal index (SI; n = 82) were measured in plants growing at their natural sites. High-altitude taxa ("high"; 2,970 m a.s.l., $pCO_2$ approx. 29 Pa in 2018, the year of sampling) included *Anthyllis vulneraria* subsp. *valesiaca* and *Arabis alpina*, whereas low-altitude taxa ("low"; 540 m a.s.l., $pCO_2$ approx. 39 Pa) comprised *Anthyllis vulneraria* subsp. *carpatica* and *Arabidopsis thaliana*. Marginal means along with their 95% confidence intervals are presented. Statistical analysis results are provided in S4 Table. Different letters denote statistically significant differences at $p < 0.05$, as determined by post hoc tests.

However, SI in low-altitude taxa (*A. thaliana* and *A. carpatica*) was influenced by both taxon and treatment, as indicated by a significant interaction term (S5 Table). While *A. carpatica* displayed only a minor and statistically non-significant response to treatment, *A. thaliana* exhibited a significantly higher SI when growing at its natural site compared to plants germinated from Col-0 seeds and cultivated at ambient $pCO_2$ in growth chambers (Fig 4D).

Additionally, plants growing in their natural environments displayed greater variability in both SD and SI compared to those cultivated under experimental conditions (Figs 4A-D).

## Response of stomatal frequency to variations in $pCO_2$

SD was significantly influenced by taxon, altitude of origin, and treatment, as indicated by a significant interaction term (S6 Table). In general, plants displayed higher SD under reduced $pCO_2$ (30 Pa) compared to ambient $pCO_2$ (42 Pa) (Fig 5A) as reflected by the significant Treatment effect (S6 Table). In *A. carpatica* and *A. alpina*, however, post hoc analyses did not detect statistically significant pairwise differences between treatments. Additionally, *A. thaliana*, a low-altitude taxon, exhibited higher SD than its high-altitude counterpart *A. alpina*, whereas no such difference was observed between the *Anthyllis* subspecies. Notably, *A. alpina* and *A. thaliana* had significantly higher SD values than the *Anthyllis* subspecies.

SI was taxon- and treatment-specific, as indicated by a significant interaction term (S6 Table). Across all taxa, SI was higher under reduced (30 Pa) compared to ambient $pCO_2$ (42 Pa), with a more pronounced effect in *Arabis* and *Arabidopsis* than in the *Anthyllis* subspecies (Fig 5B). Post hoc tests detected that the treatment effect was not statistically significant in *A. carpatica*. Furthermore, taxa responded differently depending on their altitude of origin (significant Origin × Taxon effect, S6 Table). While the low-altitude *A. carpatica* exhibited a trend toward lower SI compared to its high-altitude

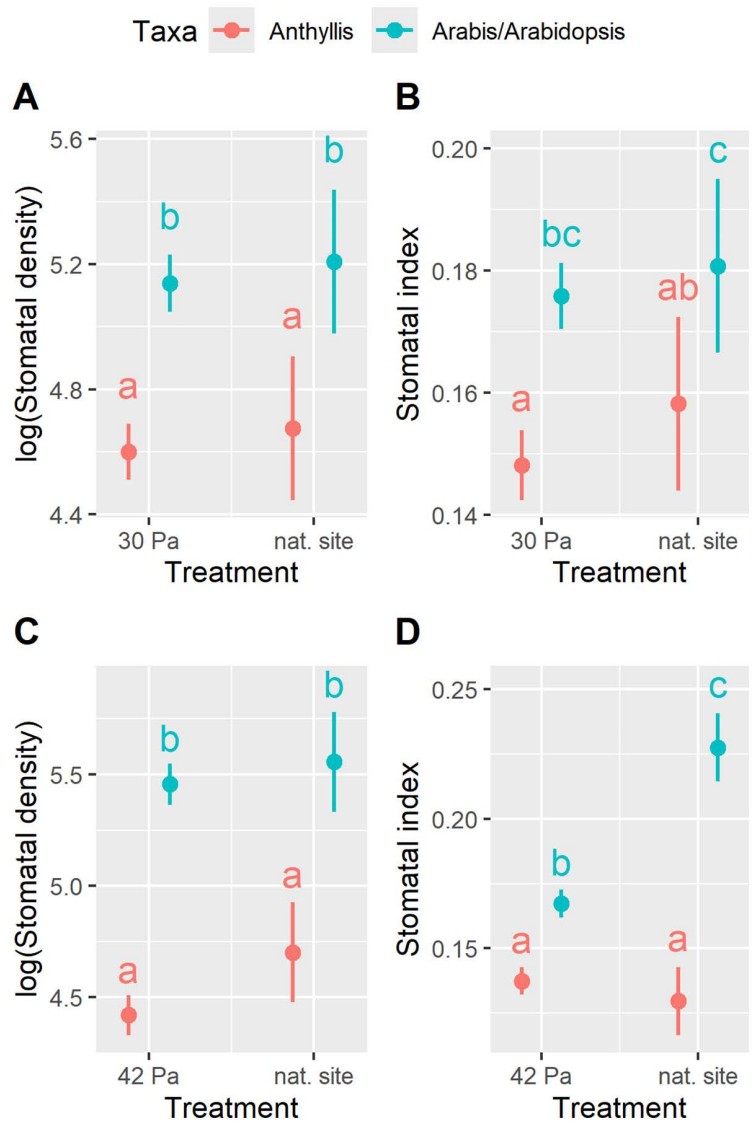

**Fig 4. Stomatal frequency of plants from natural sites and their laboratory-grown conspecifics exposed to altitude-specific pCO₂ levels.** Stomatal density (SD; A, n = 262; C, n = 239) and stomatal index (SI; B, n = 254; D, n = 239) were measured in plants growing at their natural sites (nat. site) and in those cultivated in growth chambers under altitude-specific $pCO_2$ conditions. High altitude taxa (*Anthyllis vulneraria* subsp. *valesiaca* and *Arabis alpina*) grew at 2,970 m a.s.l. ($pCO_2$ approx. 29 Pa) and were cultivated at 30 Pa $pCO_2$, while the low altitude taxa (*Anthyllis vulneraria* subsp. *carpatica* and *Arabidopsis thaliana*) grew at 540 m a.s.l. ($pCO_2$ approx. 39 Pa) and were cultivated at 42 Pa $pCO_2$. Note: *A. thaliana* cultivated in the growth chambers at 42 Pa is the Col-0 wild-type. Marginal means along with their 95% confidence intervals are presented. The results of the statistical analysis are listed in S5 Table. Different letters denote statistically significant differences at $p < 0.05$, as determined by post hoc tests.

counterpart *A. valesiaca*, the inverse was observed for *A. thaliana* and *A. alpina*. Consistent with SD results, *A. alpina* and *A. thaliana* showed higher SI values than the *Anthyllis* subspecies.

## Response of epidermal cell density to variations in pCO₂

Partial pressure of CO₂ did not have a significant effect on epidermal cell density (S2 Fig). However, epidermal cell density exhibited significant taxon- and origin-specific differences, as indicated by a significant Origin x Taxon interaction

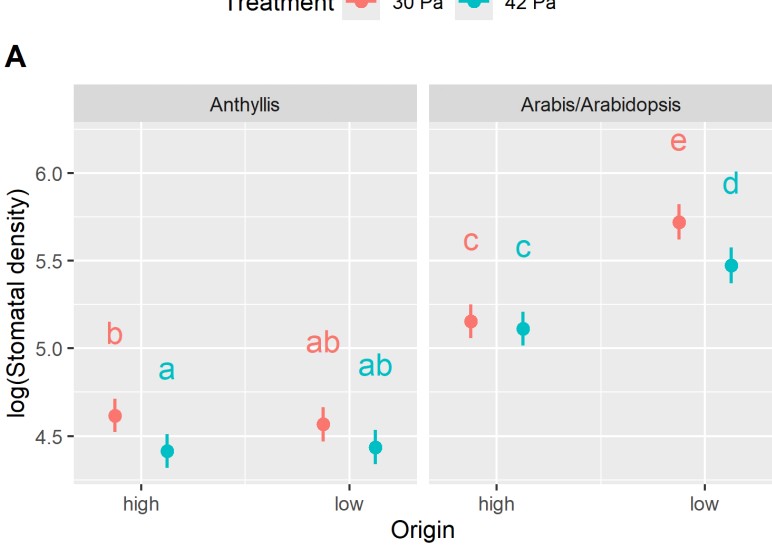

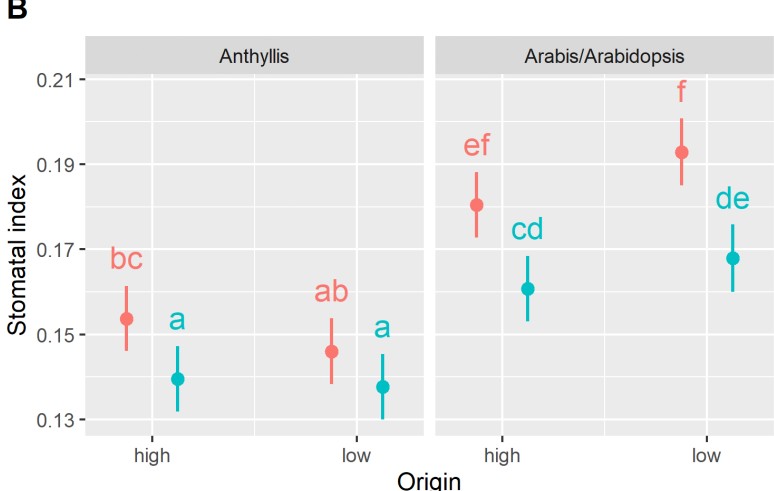

**Fig 5. Stomatal frequency (SF) response to variations in pCO₂.** Stomatal density (SD; A, n = 836) and stomatal index (SI; B, n = 828) were determined for high-altitude taxa ("high"; altitude of origin 2'970 m a.s.l.), i.e., *Anthyllis vulneraria* subsp. *valesiaca* and *Arabis alpina*, and for low-altitude taxa ("low"; 540 m a.s.l.), i.e., *Anthyllis vulneraria* subsp. *carpatica* and *Arabidopsis thaliana* (Col-0). All plants were cultivated under reduced (30 Pa) and ambient pCO₂ (42 Pa). Marginal means along with their 95% confidence intervals are presented. Statistical analysis results are provided in S6 Table. Different letters denote statistically significant differences at $p < 0.05$, as determined by post hoc tests.

(S7 Table). Both *Anthyllis* subspecies displayed similar epidermal cell densities, which were significantly lower than those observed in *A. alpina* and, in particular, *A. thaliana*.

## Response of above-ground fresh weight to variations in pCO₂

Above-ground fresh weight was significantly influenced by taxon and altitude of origin, as indicated by a significant Origin x Taxon interaction (S8 Table). The high-altitude species *A. alpina* exhibited significantly greater biomass accumulation than the low-altitude species *A. thaliana*, whereas the opposite trend was observed for the *Anthyllis* subspecies (Fig 6). In general, plants produced higher above-ground biomass under ambient pCO₂ (42 Pa) compared to reduced pCO₂ (30 Pa)

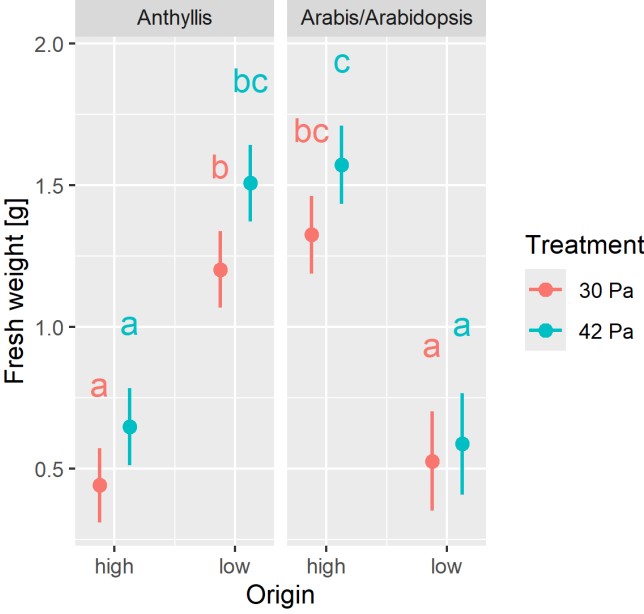

**Fig 6. Response of above-ground fresh weight to variations in pCO₂.** Fresh weight of the above-ground plant parts (n = 194) was measured in high-altitude taxa ("high"; altitude of origin 2'970 m a.s.l.), i.e., *Anthyllis vulneraria* subsp. *valesiaca* and *Arabis alpina*, and for low-altitude taxa ("low"; 540 m a.s.l.), i.e., *Anthyllis vulneraria* subsp. *carpatica* and *Arabidopsis thaliana* (Col-0). All plants were cultivated under reduced (30 Pa) and ambient pCO₂ (42 Pa). Marginal means along with their 95% confidence intervals are presented. Statistical analysis results are provided in S8 Table. Different letters denote statistically significant differences at $p < 0.05$, as determined by post hoc tests.

as reflected by a significant Treatment effect (S8 Table). However, post hoc analyses did not detect statistically significant pairwise differences between treatments.

### Response of stomatal frequency to different environmental variables

In both species, SD exhibited a negative correlation with pCO₂ (Table 2), with SD increasing as pCO₂ decreased from ambient (42 Pa) to reduced levels (30 Pa) (Fig 7A). Additionally, SD was higher at an irradiance of 175 μmol m⁻² s⁻¹ compared to 85 μmol m⁻² s⁻¹ (Fig 7D), with this effect being more pronounced in *A. thaliana* than in *A. alpina* (significant Species × Irradiance interaction, Table 2).

The effects of temperature and VPD on SD were species-specific (significant interaction terms, Table 2). In *A. thaliana*, SD increased with rising temperature and decreasing VPD, whereas *A. alpina* exhibited the opposite response to these factors (Figs 7B and 7C).

Both species exhibited a strong correlation between SI and pCO₂, as well as with irradiance (Table 3). SI increased with decreasing pCO₂ and was significantly higher at an irradiance of 175 μmol m⁻² s⁻¹ compared to 85 μmol m⁻² s⁻¹ (Figs 8A and 8D). In both species, temperature also had a significant effect on SI, with increasing temperature leading to a reduction in SI (Fig 8B).

The effect of VPD on SI was species-specific, as indicated by a significant interaction term (Table 3). While *A. alpina* exhibited a strong increase in SI with rising VPD, *A. thaliana* showed no substantial response to changes in VPD (Fig 8C).

### Discussion

The objective of this study was to analyze the stomatal responses of plants acclimated to ambient pCO₂ (low-altitude taxa) and reduced pCO₂ (high-altitude taxa) when exposed to pCO₂ variations within a range commonly encountered in natural environments, such as along altitudinal gradients.

**Table 2. Marginal effects of environmental factors on stomatal density.**

| Stomatal density (SD); n = 1,926 | Chisq | Df | Pr(>Chisq) |
|---|---|---|---|
| Species | 0.1215 | 1 | 0.7273969 |
| $pCO_2$ | 13.8340 | 1 | 0.0001997*** |
| Species × $pCO_2$ | 1.1236 | 1 | 0.2891345 |
| Temperature | 0.5205 | 1 | 0.4706273 |
| Species × Temperature | 16.4190 | 1 | 5.077e-05**** |
| VPD | 1.9677 | 1 | 0.1606885 |
| Species × VPD | 11.9219 | 1 | 0.0005548*** |
| Irradiance | 4.3951 | 1 | 0.0360428* |
| Species × Irradiance | 59.0054 | 1 | 1.572e-14*** |

Type-III ANOVA of the linear mixed model testing the statistical significance of partial pressure of $CO_2$ ($pCO_2$), air temperature, vapor pressure deficit (VPD) and irradiance as main effects for stomatal density (SD; square-root transformed) of *Arabis alpina* and *Arabidopsis thaliana* (Col-0).

Our investigations yielded the following key results: 1) Contrary to our initial hypothesis (H1), plants cultivated in the growth chambers at reduced $pCO_2$ (30 Pa) exhibited smaller stomatal apertures (SA) than those grown in the chambers at ambient $pCO_2$ (42 Pa). 2) In contrast to our second hypothesis (H2), the high-altitude taxa growing at their natural sites did not display higher stomatal frequencies (SF, i.e., stomatal density, SD, and/or stomatal index, SI) than the low-altitude taxa at their respective sites of origin. 3) Consistent with our third hypothesis (H3), plants cultivated under reduced $pCO_2$ exhibited higher SF compared to those grown under ambient $pCO_2$ conditions. 4) Above-ground biomass was reduced in the plants exposed to reduced $pCO_2$ relative to those cultivated under ambient $pCO_2$. The increased biomass under ambient $pCO_2$ was likely attributable to decreased transpiration due to lower stomatal frequency, thereby supporting our fourth hypothesis (H4).

### Response of stomatal aperture to variations in $pCO_2$

Our findings indicate that *A. thaliana* exhibited an increase in SA with decreasing $pCO_2$, whereas the other three taxa analyzed showed a tendency to close their stomata. The behavior of *A. thaliana* is in contrast to previous findings by Kammer et al. [50], where, in addition to *A. alpina*, *A. thaliana* also significantly reduced its SA in response to lower $pCO_2$. This suggests that a reduction in SA may represent a general response to decreasing $pCO_2$.

However, this observation does not align with our initial hypothesis (H1) or with existing literature, which generally supports the notion that stomatal opening occurs in response to decreasing $CO_2$ concentrations [22–24]. Kammer et al. [50] attributed the observed stomatal narrowing at reduced $pCO_2$ to an accompanying increase in vapor pressure deficit (VPD). In the present experiments, VPD conditions were largely maintained constant, effectively excluding VPD as a contributing factor to stomatal closure. Instead, we propose that the observed reduction in SA under decreased $pCO_2$ is a consequence of increased gas diffusion. In our experimental setup, lower $pCO_2$ levels were achieved by reducing ambient air pressure, inherently enhancing gas diffusivity [32,47]. Previous studies have demonstrated that increased gas diffusion can lead to elevated $CO_2$ assimilation and transpiration rates [58]. Given that the diffusion of $H_2O$ is approximately 1.6 times greater than that of $CO_2$ due to differences in molecular weight, the observed stomatal closure in our study likely reflects a protective response aimed at minimizing excessive water loss [58]. This interpretation is consistent with the optimal stomatal theory, which posits that the primary role of stomatal regulation is to minimize water loss from the plant [13]. However, experimental evidence supporting a direct link between increased $H_2O$ diffusion or reduced air pressure and stomatal closure is limited. The only study known to us that explicitly addressed this relationship reported significant stomatal narrowing in spinach (*Spinacia oleracea* L.) when air pressure was reduced from ambient (101 kPa) to 25 kPa at a constant $pCO_2$ of 40 Pa [59].

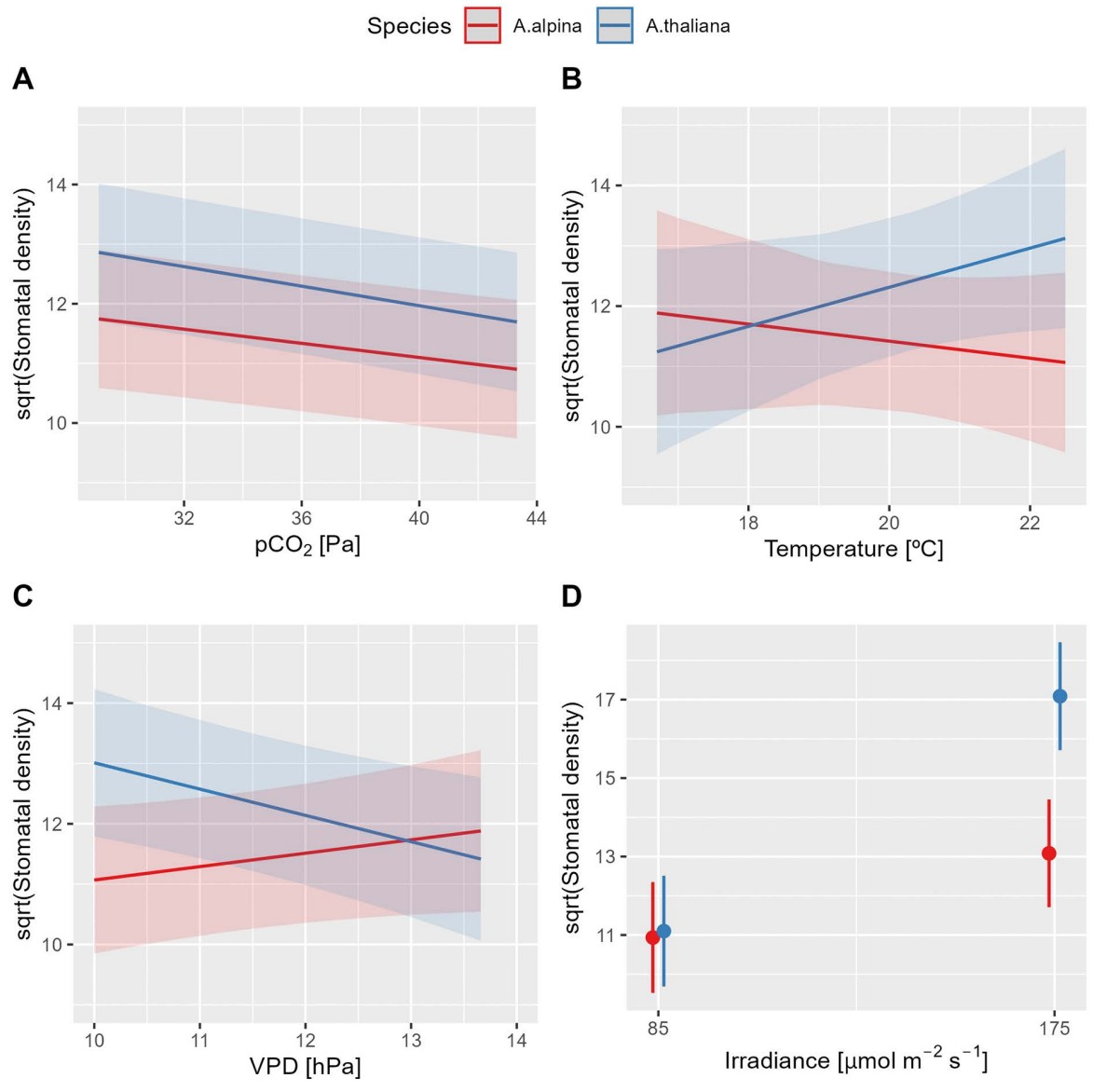

**Fig 7. Marginal effects of environmental factors on stomatal density.** The effects of (A) partial pressure of $CO_2$ (pCO₂), (B) air temperature, (C) vapor pressure deficit (VPD), and (D) irradiance on stomatal density (SD; n = 1'926) were analyzed in *Arabis alpina* and *Arabidopsis thaliana* (Col-0). The data were obtained from nine experiments conducted under environmental conditions that varied within the following minimum and maximum ranges: pCO₂ (29.1–43.3 Pa), temperature (16.7–22.5°C), VPD (10.3–13.7 hPa), and irradiance (85 and 175 µmol m⁻² s⁻¹). Marginal means along with their 95% confidence intervals are presented. Statistical analysis results are provided in Table 2.

## Response of stomatal aperture to different environmental variables

Among the environmental variables analyzed simultaneously, i.e. pCO₂, air temperature, vapor pressure deficit (VPD) and irradiance, VPD showed the strongest correlation to SA. In the experimental range of 7.7 to 13.2 hPa, both species exhibited a significant inverse response to increasing VPD, with *A. alpina* showing a stronger reduction in SA compared to *A. thaliana*. This finding aligns with our hypothesis (H5) and suggests a high sensitivity of SA to VPD, leading to stomatal narrowing even with moderate increases in VPD. The dominant role of VPD in the regulation of SA was demonstrated by

**Table 3. Marginal effects of environmental factors on stomatal index.**

| Stomatal index (SI); n = 490 | Chisq | Df | Pr(>Chisq) |
|---|---|---|---|
| Species | 0.2709 | 1 | 0.6027042 |
| pCO$_2$ | 25.6998 | 1 | 3.989e-07*** |
| Species × pCO$_2$ | 0.1312 | 1 | 0.7171946 |
| Temperature | 7.5210 | 1 | 0.0060983** |
| Species × Temperature | 2.6238 | 1 | 0.1052741 |
| VPD | 23.8381 | 1 | 1.048e-06*** |
| Species × VPD | 28.6312 | 1 | 8.756e-08*** |
| Irradiance | 11.6480 | 1 | 0.0006427*** |
| Species × Irradiance | 0.0932 | 1 | 0.7601384 |

Type-III ANOVA of the linear mixed model testing the statistical significance of partial pressure of CO$_2$ (pCO$_2$), air temperature, vapor pressure deficit (VPD) and irradiance as main effects for stomatal index (SI, square-root transformed) of *Arabis alpina* and *Arabidopsis thaliana* (Col-0).

Koolmeister et al. [60], who observed a marked reduction in the responsiveness of SA to changes in CO$_2$ levels under elevated VPD (23 hPa) compared to the normal VPD of 9 hPa.

SA showed a significant positive correlation with irradiance, which again corresponds to our hypothesis (H5). Light and CO$_2$ levels are known to be interdependent in regulating SA, with red light playing a crucial role in coordinating stomatal movements with photosynthesis [6,61,62]. Red light reduces intercellular CO$_2$ concentration due to mesophyll CO$_2$ uptake for photosynthesis [6,61]. The resulting decrease in guard cell CO$_2$ levels promotes K$^+$ uptake, leading to water influx into the guard cells and stomatal opening [61]. Consequently, maximal SA is expected under high irradiance and low CO$_2$ conditions [6], while minimal SA should occur under low irradiance and high CO$_2$ levels. A significant interaction between irradiance and pCO$_2$ was observed, indicating that a large SA due to increased irradiance does not correlate with large openings due to elevated pCO$_2$ levels. This finding aligns with the mechanism described above.

## Stomatal frequency of plants growing at their natural sites

The high-altitude taxa, which were sampled at their natural sites under conditions of low atmospheric pCO$_2$ (approximately 29 Pa in 2018, i.e., the year of sampling), did not exhibit higher SF compared to their closely related low-altitude counterparts, which grew at their natural sites with higher pCO$_2$ (approximately 39 Pa). Notably, *A. thaliana* displayed significantly higher SF than its high-altitude relative, *A. alpina*. These findings do not support hypothesis 2 (H2) and contradict prior research indicating that plants tend to increase SF with rising altitude [33–38,40–42].

Altitudinal increases in SF have been proposed as an adaptive response to compensate for lower CO$_2$ availability [39]. However, altitude-related reductions in air pressure lead to enhanced gas diffusion, although this effect is mitigated by lower temperatures. According to Terashima et al. [63] and Körner [64], increased CO$_2$ diffusion can partly counterbalance the reduced CO$_2$ availability at high altitudes, contributing to the relatively high photosynthetic capacities observed in alpine plants compared to lowland species [64,65]. However, reduced air pressure at higher elevations also leads to a lower partial pressure of water vapor, increasing transpiration due to the elevated diffusivity of H$_2$O, which is approximately 1.6 times higher than that of CO$_2$ [58]. The opposing effects of decreasing pCO$_2$ and, conversely, enhanced CO$_2$ uptake and increased transpiration – both driven by elevated diffusion rates at high altitudes – may counteract one another, potentially resulting in relatively stable SF along altitudinal gradients. Thus, the findings presented herein align with previous research indicating that SF does not exhibit a consistent altitudinal pattern [34].

 

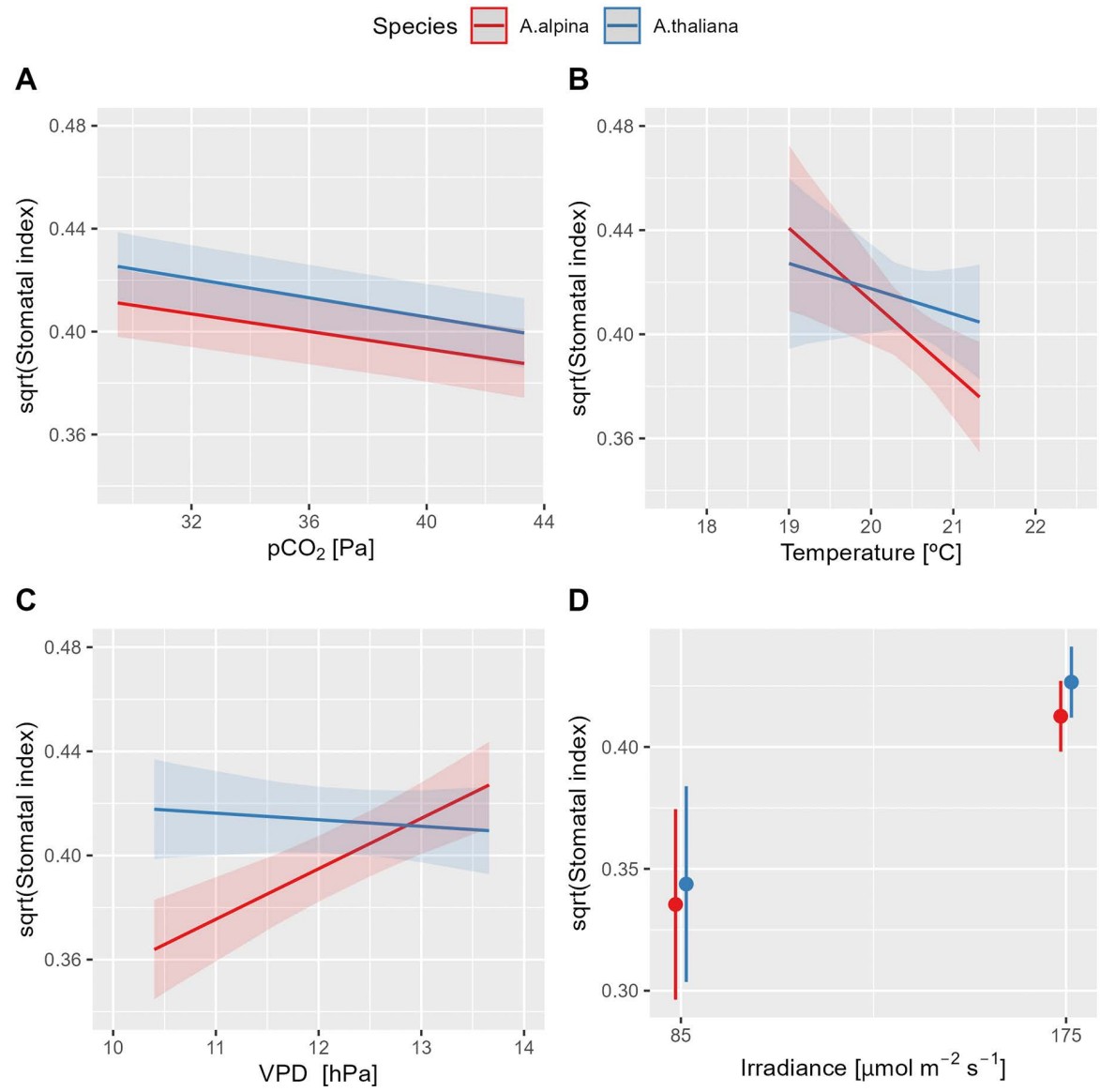

**Fig 8. Marginal effects of environmental factors on stomatal index.** The effects of (A) partial pressure of $CO_2$ ($pCO_2$), (B) air temperature, (C) vapor pressure deficit (VPD), and (D) irradiance on stomatal index (SI; n = 490) were analyzed in *Arabis alpina* and *Arabidopsis thaliana* (Col-0). The data were obtained from six experiments conducted under environmental conditions that varied within the following minimum and maximum ranges: $pCO_2$ (29.5–43.3 Pa), temperature (19.0–21.3°C), VPD (10.4–13.7 hPa), and irradiance (85 and 175 µmol m$^{-2}$ s$^{-1}$). Marginal means along with their 95% confidence intervals are presented. Statistical analysis results are provided in Table 3.

## Stomatal frequency of plants from natural sites and their laboratory-grown conspecifics exposed to altitude-specific $pCO_2$ levels

With the exception of SI in *A. thaliana*, no significant differences were observed in SF between plants growing at their natural sites and conspecifics cultivated under $pCO_2$ conditions corresponding to their original altitude. This finding suggests that the experimental conditions closely mirrored the natural growth conditions of these plants. Notably, it indicates that the relatively low irradiance applied in the experiments did not lead to a reduction in SF in the growth chamber plants compared to those at natural sites experiencing higher irradiance levels, at least temporarily.

In contrast to the *Anthyllis* subspecies and *A. alpina*, which originated from seeds collected at their natural sites, the *A. thaliana* plants used in the growth chamber experiments were obtained from a laboratory-maintained Columbia (Col-0) wild-type lineage. Under growth chamber conditions at 42 Pa $CO_2$, SI of these plants was significantly lower than that of *A. thaliana* individuals collected from natural populations. With a mean SI of 0.223 (± 0.038 SE; n = 20), the plants from natural sites had comparable values to *A. thaliana* ecotypes (mean SI ≈ 0.23; n = 75), which originated from an altitude of 790 m and were cultivated under 400 ppm $CO_2$ [49]. In contrast, the SI of Col-0 wild-type plants cultivated at ambient $pCO_2$ (42 Pa) was considerably lower, with a mean value of 0.168 (± 0.027 SE; n = 95), compared to values reported in the literature. For instance, Teng et al. [66] found an SI of approximately 0.31 at 370 ppm $CO_2$ and Tanaka et al. [67] observed an SI of approximately 0.30 at 380 ppm $CO_2$.

Initially, we hypothesized that the relatively low SF observed in the Col-0 plants in our previous experiments was an artefact of the relatively low light intensity applied, i.e., 85 μmol m$^{-2}$ s$^{-1}$ [50]. However, Yoo et al. [68] and Tanaka et al. [67] reported higher SI values of 0.24 and 0.30, respectively, under even lower irradiance conditions of approximately 100 μmol m$^{-2}$ s$^{-1}$, compared to the 175 μmol m$^{-2}$ s$^{-1}$ applied in the present study. This suggests that the Col-0 wild-type lineage used in this study is characterized by exceptionally small epidermal cells, leading to a high epidermal cell density (see S2 Fig) and an extraordinarily low SI despite a "normal" SD.

## Response of stomatal frequency to variations in $pCO_2$

Across all taxa, SD exhibited an inverse relationship with $pCO_2$, whereby a decrease in $pCO_2$ resulted in an increase in SD, and vice versa. In *A. carpatica* and *A. alpina*, however, post hoc analyses did not detect statistically significant SD responses to changes in $pCO_2$. Nevertheless, when considering the combined effects of $pCO_2$, temperature, VPD, and irradiance on SD, *A. alpina*, similar to *A. thaliana*, demonstrated a significant increase in SD with decreasing $pCO_2$. These findings indicate that the SD of the examined taxa exhibited an inverse response to $pCO_2$, irrespective of whether their original populations were acclimated to ambient $pCO_2$ (low-altitude taxa) or reduced $pCO_2$ (high-altitude taxa). Since epidermal cell density did not change significantly in response to variations in $pCO_2$, the observed response of SD to $pCO_2$ can be attributed to changes in stomatal number rather than alterations in epidermal cell size. Consequently, in the taxa investigated, a greater proportion of protodermal cells differentiated into guard cells at 30 Pa $pCO_2$ compared to 42 Pa. As a result, SI across all four taxa demonstrated an inverse response to changes in $pCO_2$.

Despite the reduction in SF at ambient $pCO_2$ (42 Pa), plants grown at 42 Pa $pCO_2$ exhibited greater above-ground fresh weight than those cultivated at 30 Pa. Given that the experimental light intensity was 175 μmol m$^{-2}$ s$^{-1}$, photosynthesis was light-limited rather than $CO_2$-limited [69], variations in $CO_2$ uptake due to changes in $pCO_2$ and SD likely did not affect photosynthetic performance [68]. Instead, the increased biomass at 42 Pa $pCO_2$ can be attributed to reduced transpirational water loss associated with lower SD [67,68]. These results support both hypotheses three (H3) and four (H4) and align with the optimal stomatal theory, which postulates that plants regulate stomatal conductance to maximize photosynthesis while minimizing water loss, thereby achieving optimal water-use efficiency [7,13,14].

Consistent with our findings, Lembo et al. [47] demonstrated that reductions in $pCO_2$, resulting from a decrease in atmospheric pressure (from 85 to 62 kPa; corresponding to altitudes of 1,500 and 4,000 m), led to increased stomatal conductance and a concurrent decrease in above-ground biomass in *Trifolium pratense* and *Hieracium pilosella*. This suggests that biomass production at 85 kPa was favored due to reduced water loss originating from decreased stomatal conductance.

While Ainsworth and Rogers [10] emphasized that stomatal conductance responses to elevated [$CO_2$] are primarily regulated by changes in SA, in the present study, reductions in stomatal conductance were solely driven by decreases in SF, with SA increasing at elevated $pCO_2$. In the study by Lembo et al. [47], no explicit distinction was made between conductance contributions from SA or SF.

The contrasting responses of SA (positive) and SF (negative) to variations in $pCO_2$ suggest the involvement of distinct $CO_2$ signaling pathways governing stomatal movement and stomatal development [24].

### Response of stomatal frequency to different environmental variables

Among the environmental variables analyzed simultaneously, including $pCO_2$, temperature, VPD, and irradiance, $pCO_2$ exerted the most pronounced effect on SF of both species, *A. alpina* and *A. thaliana*. Additionally, both species exhibited higher SF at an irradiance of 175 mol m$^{-2}$ s$^{-1}$ compared to 85 mol m$^{-2}$ s$^{-1}$, which is in line with our hypothesis (H6). Under increased irradiance, a greater supply of ATP and NADPH is generated through the light reactions of photosynthesis, and the increase in SD enhances $CO_2$ availability at carboxylation sites [12,69,70]. Together, these processes promote the carbon reactions of photosynthesis (Calvin-Benson cycle) and carbohydrate synthesis.

However, the effects of temperature and VPD on SF were less significant, inconsistent, and species-specific. This is likely due to the fluctuating nature of temperature and VPD during the experiments, meaning that their mean values may not accurately reflect the conditions experienced by a developing leaf at the time of stomata formation. In addition, the gradients of these factors were relatively short and within a moderate range, i.e., the temperature and VPD conditions were rather favorable for plant growth, and changes within these ranges may not have been capable of causing significant changes in SF.

### Origin specificity of stomatal responses

SA was the only parameter significantly affected by plant origin. Under equal $pCO_2$, the high-altitude *A. alpina* exhibited significantly reduced stomatal opening compared to the low-altitude *A. thaliana*. A similar trend was observed in the *Anthyllis* subspecies. The consistently smaller SA in high-altitude taxa may represent an adaptive response to reduced atmospheric pressure at high elevations, which enhances gas diffusion, thereby facilitating $CO_2$ uptake while simultaneously increasing transpiration rates [58]. Apart from SA, no consistent differences in stomatal responses to environmental changes were detected between low- and high-altitude taxa.

Despite the differing $pCO_2$ acclimation conditions of the two subspecies *A. carpatica* and *A. valesiaca*, both exhibited comparable SF values and largely identical responses to variations in $pCO_2$. This suggests that stomatal development is predominantly regulated by genetic determinants [19,26].

### Conclusions

The low-altitude taxa acclimated at a $CO_2$ partial pressure of approximately 39 Pa exhibited increased stomatal frequency (SF, i.e., stomatal density, SD, and stomatal index, SI) at reduced $pCO_2$ (30 Pa) compared to ambient $pCO_2$ (42 Pa). The high-altitude taxa acclimated at a $pCO_2$ of about 29 Pa displayed lower SF at ambient $pCO_2$ (42 Pa) than at reduced $pCO_2$ (30 Pa). Thus, regardless of the $pCO_2$ levels to which they were acclimated, the studied taxa exhibited an inverse response of SF to $pCO_2$ variations when other factors potentially influencing SF were held constant. Even when additional environmental variables – such as air temperature, vapor pressure deficit (VPD), and irradiance – were analyzed alongside $pCO_2$, the latter showed the most pronounced and inverse effect on SF. These findings, which show that plants acclimated to low $CO_2$ levels for centuries respond to an increase in $CO_2$ levels in the same way as plants acclimated to significantly higher $CO_2$ levels, namely with a decrease in SF, supports the hypothesis that the inverse response of SF to variations in atmospheric $CO_2$ levels is a general response pattern in $C_3$ plants.

Plants cultivated at 42 Pa $pCO_2$ produced greater above-ground biomass than those cultivated at 30 Pa. The increased biomass is likely attributable to reduced transpirational water loss due to lower stomatal conductance, which resulted from the lower SF observed at 42 Pa $pCO_2$. This finding supports the optimal stomatal theory, which postulates that plants regulate stomatal conductance to maximize photosynthesis while minimizing water loss, thereby achieving optimal water-use efficiency [7,13,14].

 

The high-altitude taxa growing at their natural sites did not exhibit higher SF than the low-altitude taxa at their respective sites of origin. This suggests that, under specific environmental conditions, factors other than atmospheric $CO_2$ levels may play a decisive role in determining SF [37,46,49]. This may also explain why only a minority of cases in meta-studies [12,27] exhibited an inverse response of SF to $CO_2$.

Overall, the results confirm the inverse response of SF to variations in atmospheric $CO_2$ levels between 300 and 420 ppm and are consistent with the predictions of optimal stomatal theory. Extrapolating these findings to future atmospheric $CO_2$ levels exceeding 420 ppm, they would suggest that plants may further reduce SF and stomatal conductance, thereby benefiting from decreased water loss. Consequently, plants in dry environments may experience the greatest advantages from increased atmospheric $CO_2$ levels. However, Purcell et al. [15] demonstrated that plants from arid and warm environments specifically exhibited an increase in stomatal conductance in response to elevated $CO_2$ levels, indicating that other environmental factors may override the effect of increased atmospheric $CO_2$ on stomatal conductance. Therefore, the relative influence of a specific environmental signal on stomatal conductance compared to other signals warrants further investigation [7].

This study focused on the stomatal responses of relatively short-lived $C_3$ dicotyledonous species, raising the question of whether similar responses to variations in atmospheric $CO_2$ levels would also occur in monocotyledons, $C_4$ species, or long-lived plants such as trees. Additionally, there are substantial doubts regarding whether plants exhibit similar stomatal responses to changes in $CO_2$ levels above 400 ppm as they do below 400 ppm [7,25,26,29,30]. Therefore, future research should investigate how stomatal conductance responds to $CO_2$ levels exceeding 400 ppm under varying temperature and humidity conditions across a broad range of plant species.

## Supporting information

**S1 Fig. Altitudinal distribution of the taxa under study.** The altitudinal distribution of the examined taxa is based on records from the National Data and Information Center on the Swiss Flora (www.infoflora.ch). *Anthyllis vulneraria* subsp. *carpatica* (A. carpatica): $n = 3,044$ observations; *Anthyllis vulneraria* subsp. *valesiaca* (A. valesiaca): $n = 1,366$ observations; *Arabidopsis thaliana* (A. thaliana): $n = 883$ observations; *Arabis alpina* (A. alpina): $n = 1,078$ observations.
(PDF)

**S2 Fig. Response of epidermal cell density to variations in $pCO_2$.** Epidermal cell density ($n = 828$) was measured in high-altitude taxa (*Anthyllis vulneraria* subsp. *valesiaca* and *Arabis alpina*, altitude of origin 2,970 m a.s.l.) and low-altitude taxa (*Anthyllis vulneraria* subsp. *carpatica*, 540 m a.s.l., and *Arabidopsis thaliana* Col-0). Plants were cultivated under reduced (30 Pa) and ambient $pCO_2$ (42 Pa). Marginal means along with their 95% confidence intervals are presented. Statistical analysis results are provided in S7 Table.
(PDF)

**S1 Table. Sites of collection of seeds and leaves for stomatal frequency analysis.** All collection sites are located in Switzerland (BE: Canton of Berne; VS: Canton of Valais). The areas entered for the sampling of leaves and seeds are not privately owned and are freely accessible. The taxa sampled are not protected by law.
(PDF)

**S2 Table. Composition of the nutrient solution used in hydroponic plant cultivation.**
(PDF)

**S3 Table. Stomatal aperture response to variations in $pCO_2$.** Type-I ANOVA of the linear mixed model testing the impact of Origin, Taxon and $pCO_2$ treatment on stomatal aperture (SA).
(PDF)

**S4 Table. Stomatal frequency of plants growing at their natural sites.** Type-I ANOVA of the linear model testing the impact of Origin and Taxon on stomatal density (SD; log-transformed) and stomatal index (SI) of plants growing at their natural sites.
(PDF)

**S5 Table. Stomatal frequency of plants from natural sites and their laboratory-grown conspecifics exposed to altitude-specific $pCO_2$ levels.** Type-I ANOVA of the linear model testing for differences in stomatal density (SD; log-transformed) and stomatal index (SI) between natural populations and their corresponding $pCO_2$ treatment in the growth chamber (Treatment) for each Taxon. A $CO_2$ partial pressure of 30 Pa corresponds to the high-altitude population while 42 Pa corresponds to the low-altitude population.
(PDF)

**S6 Table. Stomatal frequency response to variations in $pCO_2$.** Type-I ANOVA of the linear mixed model testing the impact of Origin, Taxon and $pCO_2$ treatment on stomatal density (SD; log-transformed) and stomatal index (SI).
(PDF)

**S7 Table. Response of epidermal cell density to variations in $pCO_2$.** Type-I ANOVA of the linear mixed model testing the impact of Origin, Taxon and $pCO_2$ treatment on epidermal cell density (log-transformed).
(PDF)

**S8 Table. Response of above-ground fresh weight to variations in $pCO_2$.** Type-I ANOVA of the linear mixed model testing the impact of Origin, Taxon and $pCO_2$ treatment on above-ground fresh weight.
(PDF)

## Acknowledgments

We thank Daniel Braun and Moritz Kammer for their assistance in the laboratory. We are also grateful to Joel Adler, Matthias Bigler, Lucien Bovet, Oliver Schulz, and Olivia Stowasser for their helpful and stimulating discussions during various stages of the study. We would like to thank the reviewers for their careful review of the manuscript and their valuable suggestions for improvement.

## Author contributions

**Conceptualization:** Peter Manuel Kammer, Christian Schöb.

**Data curation:** Peter Manuel Kammer, Dominik Lukas Wermelinger, Janick Michael Klossner, Jonathan Simon Steiner.

**Formal analysis:** Dominik Lukas Wermelinger, Janick Michael Klossner, Christian Schöb.

**Investigation:** Peter Manuel Kammer, Dominik Lukas Wermelinger, Janick Michael Klossner, Jonathan Simon Steiner.

**Methodology:** Peter Manuel Kammer, Christian Schöb.

**Project administration:** Peter Manuel Kammer.

**Resources:** Peter Manuel Kammer.

**Supervision:** Peter Manuel Kammer.

**Validation:** Peter Manuel Kammer, Christian Schöb.

**Visualization:** Peter Manuel Kammer, Christian Schöb.

**Writing – original draft:** Peter Manuel Kammer, Christian Schöb.

**Writing – review & editing:** Peter Manuel Kammer, Dominik Lukas Wermelinger, Janick Michael Klossner, Jonathan Simon Steiner, Christian Schöb.

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
