## [Decision Letter · Decision Letter 0]

9 Oct 2025

Dear Dr. Kammer,

Thank you for submitting your manuscript to PLOS ONE. After careful consideration, we feel that it has merit but does not fully meet PLOS ONE’s publication criteria as it currently stands. Therefore, we invite you to submit a revised version of the manuscript that addresses the points raised during the review process.

We look forward to receiving your revised manuscript.

Kind regards,

Mayank Anand Gururani

Academic Editor

PLOS ONE

Journal Requirements:

Reviewers' comments:

Reviewer's Responses to Questions

**Comments to the Author**

1. Is the manuscript technically sound, and do the data support the conclusions?

Reviewer #1: Yes

Reviewer #2: Yes

2. Has the statistical analysis been performed appropriately and rigorously?

Reviewer #1: Yes

Reviewer #2: Yes

3. Have the authors made all data underlying the findings in their manuscript fully available?

Reviewer #1: Yes

Reviewer #2: Yes

4. Is the manuscript presented in an intelligible fashion and written in standard English?

Reviewer #1: Yes

Reviewer #2: Yes

Reviewer #1: Adaptation of plants to environmental conditions, in particular to altitude ranges, is important for understanding the adaptation of photosynthesis processes. The results confirm the regulation of stomata conduction in response to changes in the amount of CO2. It is possible to make a forecast of changes in the conductivity of stomata with an increase in the level of CO2 in the atmosphere. However, the question arises whether all plants will react equally to the CO2 content. There may be a need for research on other plants.

Reviewer #2: I found this manuscript interesting to read and fairly well-written. However, there are some issues that the authors need to address before the manuscript could be accepted for publication. Detailed comments are given below.

Abstract

L19: “Understanding the stomatal responses …” instead of “Comprehension of the stomatal …”.

L21: remove “, for example”.

LL22-25: not very clear to read. Please revise to improve clarity as to what treatments were used and how the treatments were applied (under controlled lab experiments).

L27: “increasing Co2 levels ..” instead of the current.

L32: “2,970 m a.s.l” instead of “2’970 m a.s.l”.

L36: local environmental conditions like what? Give examples.

Introduction

L45: “into the atmosphere” instead of “in the atmosphere”.

LL50-51: the reference should be placed at the end of the sentence, not the in the middle.

L51: better to use Co2 levels throughout the manuscript instead of [Co2].

LL51-52: continuously increasing atmospheric Co2 due to what?

L54: remove “, for example”.

L60: “periods of drought” instead of “drought periods”.

LL63-66: the terminology adopted here should be referenced.

LL66-67: needs a reference.

LL69-71: needs a reference.

L74: field or lab experiments?

L75: “an inverse relationship with Co2 levels” instead of the current.

LL77-79: were those findings documented under lab or field experiments, or both?

LL80-81: “the present-day concentrations” instead of the current.

L82: comma after “[26]”.

LL90-92: responses of what is being discussed here?

LL107-111: “ .. we tested plants originating from populations that have grown for prolonged …” instead of the current.

LL107-111: the language here needs be streamlined and the sentences need to be simplified a bit.

L114: please add” to minimize the confounding effects of those environmental factors in the experiment” at the end of the sentence.

LL117-119: the terminology adopted here should be referenced.

L123: “grown” instead of cultivated.

Materials and Methods

L140: “have been growing” instead of “have been living”.

LL148-152: references are needed.

LL155-157: unclear to me if seeds were collected from plants and grown in growth chambers for experiments or if seeds were obtained from the Col-0 wild type, please clarify which seeds were used in the experiments. Also, why seeds of the Col-0 wild type were used?

L158: “could be” instead of “can be”.

LL186-188: but those were lab conditions not field conditions, as the growth chambers were placed in a lab. How would the fluctuation of temperature and humidity under lab conditions be similar to their fluctuation under field conditions?

L189: “grown” instead of “cultivated”.

LL192-194: why the plants in the first group were grown for 28 days and for 35 days in the second group? Explain to the reader.

L195: “to prevent any damage to the root system” instead of the current.

L199: “Measurements of Plant Traits” instead of the current.

L200: “For measuring stomatal aperture …” instead of the current.

L206: “For measuring stomatal frequency …” instead of the current.

LL209-217: relevant references should be cited here.

L229: “after the end of the experiment.” instead of the current.

Statistical Analysis

L234: “as the main effects” instead of “as explanatory variables”.

L235: why chamber ID was used as a random effect?

L236: those are “liner models” not regression analysis. Please make sure you use the correct terminology throughout the manuscript.

L247: “as main effects” instead of “as explanatory variables”.

LL249-250: is there a refence to support this approach?

L253: “as main effects” instead of “as explanatory variables”.

LL257-258: “as main effects” instead of “as explanatory variables”.

LL263-264: I did not see any co-variate analysis in the manuscript. So how could that be? Was there bias in the experiment? If yes, this is not the right way to reduce bias in the experiments. Please explain.

LL266-273: seems to align with the policy of the Journal, yet I do not like to see AI tools being used for generating or composing texts in scientific papers.

Results

L278: why is the stats table placed in the appendix? They should be a core part of the manuscript.

LL292-298: As I could see in Figure 2, the responses to temperature and pCo2 were also analyzed but not mentioned here. Why is that?

L321: why is the stats table placed in the appendix?

L335: “grown” instead of “cultivated”.

L340: why is the stats table placed in the appendix?

LL357-358: which figures show this trend?

L362: why is the stats table placed in the appendix?

L387: why are the figures placed in the appendix?

L389: why is the stats table placed in the appendix?

L395: why is the stats table placed in the appendix?

Tables

Table 1: “n=1,200” instead of “1’200”.

L303: “main effects”.

Why all degrees of freedom are 1 even in interactions?

Table 2: “n=1,926” instead of “1’926”.

Why all degrees of freedom are 1 even in interactions?

Table 3: Why all degrees of freedom are 1 even in interactions?

Discussion

L495: “diffusion” instead of “diffusivity”.

L552: “grown” instead of “cultivated”.

When the authors say, “plant grew under natural conditions,” I am not sure what they mean by that. Plants were grown in growth chambers, so how does that relate to natural conditions?

LL560-568: so we have leaves collected from natural sites and leaves collected from plants grown in growth chambers, which ones were included in the analysis? The authors did not describe their methods well to clarify this, which makes the Discussion confusing for the reader. The authors need to address this and improve the clarity of their Methods and Discussion.

Conclusions

OK.

Figures

Figure 1: The plant species under study should be specified in a better way, the reader does not need to figure out which is which on the figure. Also, the figures should show any significant differences between/within treatments, or lack thereof.

Figure 2: The plant species under study should be specified in a better way. The figures should show any significant differences between/within treatments, or lack thereof. L308: : “n=1,200” instead of “1’200”.

Figure 3: The plant species under study should be specified in a better way. Also, the figures should show any significant differences between/within treatments, or lack thereof.

Figure 4: The plant species under study should be specified in a better way. Also, the figures should show any significant differences between/within treatments, or lack thereof.

Figure 5: The plant species under study should be specified in a better way. Also, the figures should show any significant differences between/within treatments, or lack thereof.

Figure 6: The plant species under study should be specified in a better way. Also, the figures should show any significant differences between/within treatments, or lack thereof.

Figure 7: The plant species under study should be specified in a better way. Also, the figures should show any significant differences between/within treatments, or lack thereof.

Figure 8: The plant species under study should be specified in a better way. Also, the figures should show any significant differences between/within treatments, or lack thereof.

.

Reviewer #1: **Yes:** Plant physiologist Dr. Oksana BelousPlant physiologist Dr. Oksana BelousPlant physiologist Dr. Oksana BelousPlant physiologist Dr. Oksana Belous

Reviewer #2: No

---

## [Author Response · Author response to Decision Letter 1]

1 Dec 2025

Response to Reviewers (PONE-D-25-38428)

We would like to thank the reviewers for their careful examination of our manuscript and their very valuable suggestions for improvement. We are confident that these improvements will enhance the quality of the article.

In the following, we will address each individual point raised by the reviewers.

Author’s replies to the comments of Reviewer 1

… However, the question arises whether all plants will react equally to the CO2 content. …

We have now addressed this question at the end of the “Conclusions” section.

Author’s replies to the comments of Reviewer 2

Reviewer #2: I found this manuscript interesting to read and fairly well-written. However, there are some issues that the authors need to address before the manuscript could be accepted for publication. Detailed comments are given below.

Abstract

L19: “Understanding the stomatal responses …” instead of “Comprehension of the stomatal …”.

We have changed this word.

L21: remove “, for example”.

Ok, removed.

LL22-25: not very clear to read. Please revise to improve clarity as to what treatments were used and how the treatments were applied (under controlled lab experiments).

We have revised this section to improve readability and comprehensibility.

L27: “increasing Co2 levels ..” instead of the current.

Ok, changed.

L32: “2,970 m a.s.l” instead of “2’970 m a.s.l”.

Ok, changed.

L36: local environmental conditions like what? Give examples.

We added “low humidity or elevated temperatures” as such specific local conditions.

Introduction

L45: “into the atmosphere” instead of “in the atmosphere”.

Ok, changed.

LL50-51: the reference should be placed at the end of the sentence, not the in the middle.

Reference was placed at the end of the sentence.

L51: better to use Co2 levels throughout the manuscript instead of [Co2].

We now have written “CO2 levels” throughout the manuscript.

LL51-52: continuously increasing atmospheric Co2 due to what?

We added the cause of this increase.

L54: remove “, for example”.

Ok, removed.

L60: “periods of drought” instead of “drought periods”.

Ok, changed.

LL63-66: the terminology adopted here should be referenced.

We added three references: Willmer & Fricker 1996, Franks & Beerling 2009, Zhang et al. 2012

LL66-67: needs a reference.

We added three references: Ainsworth & Rogers 2007, Roelfsema & Kollist 2013, Xu et al. 2016

LL69-71: needs a reference.

We added three references: Ainsworth & Rogers 2007, Roelfsema & Kollist 2013, Xu et al. 2016

L74: field or lab experiments?

We have rewritten this section and added a sentence to explain this point.

L75: “an inverse relationship with Co2 levels” instead of the current.

Ok, changed.

LL77-79: were those findings documented under lab or field experiments, or both?

We have rewritten this section and added a sentence to explain this point.

LL80-81: “the present-day concentrations” instead of the current.

Ok, changed.

L82: comma after “[26]”.

We have put the comma in place.

LL90-92: responses of what is being discussed here?

We have specified the response.

LL107-111: “ .. we tested plants originating from populations that have grown for prolonged …” instead of the current.

Ok, changed.

LL107-111: the language here needs be streamlined and the sentences need to be simplified a bit.

We have revised this section to improve readability and comprehensibility.

L114: please add” to minimize the confounding effects of those environmental factors in the experiment” at the end of the sentence.

We have added this sentence.

LL117-119: the terminology adopted here should be referenced.

We added two references: Willmer & Fricker 1996, Yan et al. 2017

L123: “grown” instead of cultivated.

Ok, changed.

Materials and Methods

L140: “have been growing” instead of “have been living”.

Ok, changed.

LL148-152: references are needed.

We have added the reference for this data.

LL155-157: unclear to me if seeds were collected from plants and grown in growth chambers for experiments or if seeds were obtained from the Col-0 wild type, please clarify which seeds were used in the experiments. Also, why seeds of the Col-0 wild type were used?

We reworded this section to clarify this point.

L158: “could be” instead of “can be”.

Ok, changed.

LL186-188: but those were lab conditions not field conditions, as the growth chambers were placed in a lab. How would the fluctuation of temperature and humidity under lab conditions be similar to their fluctuation under field conditions?

We agree that this statement is rather confusing, and have deleted the last part of the sentence.

L189: “grown” instead of “cultivated”.

Ok, changed.

LL192-194: why the plants in the first group were grown for 28 days and for 35 days in the second group? Explain to the reader.

We have now explained this point.

L195: “to prevent any damage to the root system” instead of the current.

Ok, changed.

L199: “Measurements of Plant Traits” instead of the current.

Ok, changed.

L200: “For measuring stomatal aperture …” instead of the current.

Ok, changed.

L206: “For measuring stomatal frequency …” instead of the current.

Ok, changed.

LL209-217: relevant references should be cited here.

We added two references: Weyers & Johansen 1985, Lawson et al. 1998

L229: “after the end of the experiment.” instead of the current.

Ok. Changed.

Statistical Analysis

L234: “as the main effects” instead of “as explanatory variables”.

We have changed this term throughout the manuscript.

L235: why chamber ID was used as a random effect?

There were two chambers that were interchangeably used for the different air pressure treatments. By including chamber ID as a random effect, we wanted to control for chamber-specific effects, which were unlikely to occur, but cannot be excluded. Chambers in this case can be considered similar to blocks. We have added a few words to explain this point.

L236: those are “liner models” not regression analysis. Please make sure you use the correct terminology throughout the manuscript.

We have changed this term throughout the manuscript.

L247: “as main effects” instead of “as explanatory variables”.

Ok, changed.

LL249-250: is there a refence to support this approach?

This procedure was used to increase experimental precision and reduce effects of nuisance variables. In this case, we wanted to control for possible, though unlikely, differences among positions within a chamber. The purpose is the same as with chamber ID as random term, i.e. controlling for potential confounding factors due to the experimental setup. This is a common practice in experimental statistics; however, we cannot cite a specific reference for this particular approach.

L253: “as main effects” instead of “as explanatory variables”.

Ok, changed.

LL257-258: “as main effects” instead of “as explanatory variables”.

Ok, changed.

LL263-264: I did not see any co-variate analysis in the manuscript. So how could that be? Was there bias in the experiment? If yes, this is not the right way to reduce bias in the experiments. Please explain.

We tested for observer bias by including the observer ID, i.e. the person taking the observations, as a covariate in the initial analyses. However, as this covariate was insignificant in all cases, we did not report them in the final analyses. We added a sentence to clarify this point.

Use of artificial intelligence tools

LL266-273: seems to align with the policy of the Journal, yet I do not like to see AI tools being used for generating or composing texts in scientific papers.

We fundamentally agree with this statement. However, AI tools can be useful for non-native speakers to improve the text in terms of readability and clarity.

Results

L278, 321, 340, 362, 389, 395: why is the stats table placed in the appendix? They should be a core part of the manuscript.

We decided to present the ANOVA tables of the statistical analyses in the Appendix (Supporting information) as they simply support the results visualized in the corresponding figures shown in the main text. As all our factors have 1 degree of freedom, their significance can easily be assessed with the figure. However, if the editor wishes to move the ANOVA tables from the Appendix to the main text, we are happy to do so.

LL292-298: As I could see in Figure 2, the responses to temperature and pCo2 were also analyzed but not mentioned here. Why is that?

We added a sentence saying that the response of stomatal aperture to temperature and pCO2 was not significant.

L335: “grown” instead of “cultivated”.

Ok, changed.

LL357-358: which figures show this trend?

Figs 4A-D. We have added the corresponding reference.

L387: why are the figures placed in the appendix?

We decided to present the figure depicting the response of epidermal cell density to pCO2 in the Appendix (Supporting information) as it is not a central finding of our study. Epidermal cell density was determined to calculate the stomatal index. However, if the editor wishes to move this figure from the Appendix to the main text, we are happy to do so.

Tables

Table 1: “n=1,200” instead of “1’200”.

Ok, changed.

L303: “main effects”.

Ok, changed.

Tables 1 to 3: Why all degrees of freedom are 1 even in interactions?

All main effects use 1 degree of freedom either because they stand for a factor with two factor levels or for a factor as continuous variable. And if all main effects use one degree of freedom, the interactions use only one df as well.

Table 2: “n=1,926” instead of “1’926”.

Ok, changed.

Discussion

L495: “diffusion” instead of “diffusivity”.

Ok, we have changed this term.

L552: “grown” instead of “cultivated”.

Ok, changed.

When the authors say, “plant grew under natural conditions,” I am not sure what they mean by that. Plants were grown in growth chambers, so how does that relate to natural conditions?

We have reworded the title of this section to clarify this point.

LL560-568: so we have leaves collected from natural sites and leaves collected from plants grown in growth chambers, which ones were included in the analysis? The authors did not describe their methods well to clarify this, which makes the Discussion confusing for the reader. The authors need to address this and improve the clarity of their Methods and Discussion.

Yes, we analyzed the stomatal frequency of leaves collected from natural sites as well as leaves from plants grown in growth chambers. Most analyses were performed using leaves from plants grown in growth chambers. When leaves from natural sites were included, this is indicated in the titles of the corresponding sections in the “Results” and “Discussion” sections (“Stomatal frequency of plants growing in their natural sites” and “Stomatal frequency of plants from natural sites and their laboratory-grown conspecifics exposed to altitude-specific pCO₂ levels”) as well as in the captions of the corresponding figures (3 and 4). We have also rewritten the beginning of this paragraph (LL560-568) to further clarify this point.

Conclusions

OK.

Figures

Figure 1 to 8: The plant species under study should be specified in a better way, the reader does not need to figure out which is which on the figure. Also, the figures should show any significant differences between/within treatments, or lack thereof.

We are not sure what specification with the plant species is unclear. The genus names are mentioned clearly in the figure and the figure caption specifies the full name of each species.

Significant differences are visible since means and confidence intervals are displayed in all figures. Under these circumstances, the significance of main effects can be judged based on the level of overlap of the confidence intervals. E.g. non-overlapping confidence intervals suggest statistically significant differences, while means with confidence intervals of a treatment that overlap with means of other treatments can be considered as not significantly different. This rule of thumb is well applied in our study, as can be corroborated by the ANOVA tables in the Appendix. We therefore don’t see an added value of including symbols or letters to the figures. Letters are generally displayed for posthoc tests on factors with more than two levels, which were not used in our study. In our study we use a priori contrasts specified with our main effects with 1 degree of freedom. However, if the editor thinks that it would be appropriate and desired to display a symbol or letters to reflect statistical results in the figure, we can include them.

L308: “n=1,200” instead of “1’200”.

Ok, changed.

---

## [Decision Letter · Decision Letter 1]

26 Jan 2026

Dear Dr. Kammer,

Thank you for submitting your manuscript to PLOS ONE. After careful consideration, we feel that it has merit but does not fully meet PLOS ONE’s publication criteria as it currently stands. Therefore, we invite you to submit a revised version of the manuscript that addresses the points raised during the review process.

We look forward to receiving your revised manuscript.

Kind regards,

Raffaella Balestrini

Academic Editor

PLOS One

Journal Requirements:

**Additional Editor Comments:**

The manuscript has been improved with respect to the original version. Thanks. If possible, I suggest to try to reply to the additional comment by the reviewer to increase the reading.

Reviewers' comments:

Reviewer's Responses to Questions

**Comments to the Author**

Reviewer #2: All comments have been addressed

2. Is the manuscript technically sound, and do the data support the conclusions?

Reviewer #2: Yes

3. Has the statistical analysis been performed appropriately and rigorously?

Reviewer #2: Yes

4. Have the authors made all data underlying the findings in their manuscript fully available?

Reviewer #2: Yes

5. Is the manuscript presented in an intelligible fashion and written in standard English?

Reviewer #2: Yes

Reviewer #2: I found the revised version of the manuscript improved, well-written, and well-structured. I thank the authors for addressing my comments and concerns on the previous version. I have a few minor suggestions as below. Other than that, the manuscript is in a good shape.

Abstract

Should be one paragraph, not separated into multiple paragraphs.

Introduction

L55: “to variations in Co2 levels” instead of “to Co2 levels variations”.

Materials and Methods

L205: “nutrient solution” instead of “nutrition solution”.

Statistical Analysis

Good.

Results

Good.

Tables

Good.

Discussion

Good.

Conclusions

Good.

Figures

I still think that statistical differences between treatments and/or species or lack thereof should be shown on the figures using labels or letters. I understand that overlapping and non-overlapping error bar tell the story, but it is always better to have the figures labeled appropriately so they are very straightforward and easy to understand by the reader without referring to the text. I think the Editor will agree with me on this.

.

Reviewer #2: No

---

## [Author Response · Author response to Decision Letter 2]

11 Mar 2026

Author’s replies to the comments of Reviewer 2

Reviewer 2: Comments to the Author

Reviewer #2: I found the revised version of the manuscript improved, well-written, and well-structured. I thank the authors for addressing my comments and concerns on the previous version. I have a few minor suggestions as below. Other than that, the manuscript is in a good shape.

Abstract

Should be one paragraph, not separated into multiple paragraphs.

Ok, changed

Introduction

L55: “to variations in Co2 levels” instead of “to Co2 levels variations”.

Ok, changed

Materials and Methods

L205: “nutrient solution” instead of “nutrition solution”.

Ok, changed

Statistical Analysis

Good.

Results

Good.

Tables

Good.

Discussion

Good.

Conclusions

Good.

Figures

I still think that statistical differences between treatments and/or species or lack thereof should be shown on the figures using labels or letters. I understand that overlapping and non-overlapping error bar tell the story, but it is always better to have the figures labeled appropriately so they are very straightforward and easy to understand by the reader without referring to the text. I think the Editor will agree with me on this.

To address the reviewer's concerns, we performed post hoc tests and marked the statistically significant differences in Figures 1, 3, 4, 5, and 6 with different letters. This procedure required some additional explanations, particularly in the “Methods” and “Results” sections. We hope that the figures are now clear and easy to understand.

---

## [Editor Report · Decision Letter 2]

16 Mar 2026

Stomatal responses of differently CO2-acclimated plants to natural and experimental CO2 gradients

PONE-D-25-38428R2

Dear Dr. Kammer,

We’re pleased to inform you that your manuscript has been judged scientifically suitable for publication and will be formally accepted for publication once it meets all outstanding technical requirements.

Kind regards,

Raffaella Balestrini

Academic Editor

PLOS One
---

## [Editor Report · Acceptance letter]

PONE-D-25-38428R2

PLOS One

Dear Dr. Kammer,

I'm pleased to inform you that your manuscript has been deemed suitable for publication in PLOS One. Congratulations! Your manuscript is now being handed over to our production team.

Kind regards,

on behalf of

Dr Raffaella Balestrini

Academic Editor

PLOS One